# Using a Chemical Genetic Screen to Enhance Our Understanding of the Antimicrobial Properties of Gallium against *Escherichia coli*

**DOI:** 10.3390/genes10010034

**Published:** 2019-01-09

**Authors:** Natalie Gugala, Kate Chatfield-Reed, Raymond J. Turner, Gordon Chua

**Affiliations:** 1Department of Biological Sciences, University of Calgary, 2500 University Dr. NW, Calgary, AB T2N 1N4, Canada; ngugala@ucalgary.ca (N.G.); turnerr@ucalgary.ca (R.J.T.); 2Seidman Cancer Center, University Hospitals, 11100 Euclid Ave, Cleveland, OH 44106, USA; katherine.chatfield-reed@UHhospitals.org

**Keywords:** *Escherichia coli*, gallium, antimicrobial agents, metal toxicity, metal resistance, metal-based antimicrobials

## Abstract

The diagnostic and therapeutic agent gallium offers multiple clinical and commercial uses including the treatment of cancer and the localization of tumors, among others. Further, this metal has been proven to be an effective antimicrobial agent against a number of microbes. Despite the latter, the fundamental mechanisms of gallium action have yet to be fully identified and understood. To further the development of this antimicrobial, it is imperative that we understand the mechanisms by which gallium interacts with cells. As a result, we screened the *Escherichia coli* Keio mutant collection as a means of identifying the genes that are implicated in prolonged gallium toxicity or resistance and mapped their biological processes to their respective cellular system. We discovered that the deletion of genes functioning in response to oxidative stress, DNA or iron–sulfur cluster repair, and nucleotide biosynthesis were sensitive to gallium, while Ga resistance comprised of genes involved in iron/siderophore import, amino acid biosynthesis and cell envelope maintenance. Altogether, our explanations of these findings offer further insight into the mechanisms of gallium toxicity and resistance in *E. coli*.

## 1. Introduction

The therapeutic capabilities of gallium(III) (Ga) have been and continue to be exploited for a number of clinical applications, which include: the treatment of cancer, autoimmune and infectious diseases, for the localization of tumors, inflammation and infection sites, and the reduction of accelerated bone resorption [1,2]. At the nuclear level, certain characteristics of this abiogenic metal permit essential metal mimicry, owing its similarities to Fe. In particular, the pharmacological characteristics of Ga are likely a result of its Fe(III)-like coordination chemistry and its ability to form stable six-coordinated complexes through ionic bonding [3]. This metal is trivalent and a hard acid in solution, according to the hard-soft acid-base theory [4], binding well with strong Lewis bases. As a result, Ga tends to form bonds with oxygen predominantly forming Ga(OH)_4−_ (gallate) at pH 7.4 [5].

Despite Ga’s similarities to the essential metal Fe, these metals share two main differences: (i) Ga cannot be reduced under biologically relevant reduction potentials, whereas Fe can be readily changed to and from a reduced state; and (ii) the concentration of unbound Fe(III) in solution is extremely low, localized primarily as a neutral complex with organic compounds, whereas gallate, which is anionic, can exist at significant concentrations [6].

As an Fe(III) mimetic, Ga(III) can incorporate itself into proteins and enzymes replacing Fe and effectively halting several essential metabolic processes [7,8,9,10,11,12,13,14]. Since the bioavailability of Fe is scarce, organisms, such as bacteria, have produced a variety of biomolecular chelating scavenging systems including siderophores and Fe-chelating proteins. Cells rapidly multiplying are more susceptible to Ga toxicity due to their high Fe demands [0]. As a result, this metal is approved by the US Food and Drug Administration for the treatment of cancer-associated hypercalcemia (Ganite^®^, Genta, NJ, USA) and has been tested as an antimicrobial agent against a variety of organisms including *Mycobacterium tuberculosis* [15,16], *Pseudomonas aeruginosa* [9,10,17], *Staphylococcus aureus* [18], *Rhodococcus equi* [19], *Acinetobacter baumannii* [20], and *Escherichia coli* [21].

In general, proposed mechanisms of toxicity for metal-based antimicrobials include the production and propagation of reactive oxygen species (ROS), the disruption of Fe-sulfur centers, thiol coordination, the exchange of a catalytic or structural metal, which in turn may lead to protein dysfunction, obstructed nutrient uptake, and genotoxicity [22]. The route by which Ga enters the cells is unknown, although, it is predominantly assumed that this metal crosses the cytoplasmic membrane by exploiting Fe-uptake routes, such as siderophores [23]. Several studies have explored the use of Fe-chelators as “Trojan horses” as a means of improving the delivery and toxicity of this metal in bacterial cells [14]. Still, there is insufficient research demonstrating that complexes of Ga and Fe-chelators/siderophores, such as Ga-citrate, increase the antibacterial abilities of this metal mainly since the import of this metal is not suggested to be the limiting step [23]. Furthermore, Ga exposure has been demonstrated to trigger the production of ROS in vitro [7,8]. Upon the cytoplasmic replacement of Fe with Ga, the available Fe pool is thought to increase, in turn fostering Fenton chemistry [22].

Bacteria have developed mechanisms of resistance as a means of withstanding metal toxicity. Some mechanisms include extracellular and intracellular sequestration, efflux, reduced uptake, repair, metabolic by-pass, and chemical modification [24]. Microbial resistant mechanisms associated with Ga have been studied to a far lesser degree, nonetheless, studies have shown that Ga is not as effective as postulated. For example, Ga resistance in *P. aeruginosa* and *Burkholderia cepacia* has been identified, suggested to be the result of decreased Ga import and the formation of bacterial biofilms [25,26].

Currently, research in this field is directed toward discovering novel utilities for this metal, still, the expansion of Ga as a therapeutic antimicrobial has been delayed compared to other metal-based antimicrobials, such as silver and copper. In short, it is essential that the mechanisms of Ga action in microbes are explored to greater degree in order to further the development of this antimicrobial agent.

In this work, we hypothesized that Ga exerts toxicity on multiple targets. Furthermore, we believe that there are several mechanisms of resistance that are fundamental to an organism’s adaptive response under sub-lethal concentrations of Ga. To evaluate this, we performed a genotypic screening workflow of an *E. coli* mutant library composed of 3985 strains. Each strain contains a different inactivated non-essential gene. Genome-wide toxin/stressor-challenge workflows have been used to study silver [27,28,29,30], copper [31,32], cadmium [33], cobalt [33], and zinc [34]; however, no such study has been implemented to examine the effects of Ga. Therefore, as a means of complementing existing work, we have identified a number of genes that may be involved in Ga toxicity or resistance and mapped their biological processes to their respective cellular system in *E. coli*.

## 2. Materials and Methods

All methods are as described previously by Gugala et al. [30] and all chemicals were obtained from VWR International, Mississauga, Canada, unless otherwise stated.

### 2.1. Escherichia coli Strains

The Keio collection [35] consisting of 3985 single gene *Escherichia coli* BW25113 mutants (*lacI*^q^
*rrnB*_T14_ Δ*lacZ*_WJ19_
*hsdR*514 Δ*ara*BAD_AH33_ Δ*rha*BAD_LD78_), was obtained from the National BioResource Project *E. coli* (National Institute of Genetics, Shizuoka, Japan).

### 2.2. Determination of the Minimal Inhibitory Concentration and Controls

The sublethal inhibitory concentration, a concentration below the minimal inhibitory concentration that is found to visibly challenge selected mutants under prolonged metal exposure, was determined using Δ*recA*, Δ*lacA* and Δ*lacY* strains from the Keio collection. The protein RecA is involved in a number of processes, including homologues recombination and the induction of the SOS response in reaction to DNA damage [36]. Evidence may suggest that Ga causes the formation of ROS, although the precise mechanism of production is unknown. As a result, the absence of this gene was anticipated to confer the Ga sensitive phenotype, implied by a decrease in colony formation, since it is thought to be involved in mitigating ROS stress. Further, the protein products of *lacA* and *lacY* were not anticipated to be involved in Ga resistance or toxicity, therefore mutant strains of these genes were used as negative controls. Strains Δ*recA*, Δ*lacA,* and Δ*lacY*, and the wild-type (WT) were grown overnight at 37 °C on M9 minimal media plates (6.8 g/L Na_2_HPO_4_, 3.0 g/L KH_2_PO_4_, 1.0 g/L NH_4_Cl, 0.5 g/L NaCl, 4.0 mg/L glucose, 0.5 mg/L MgSO_4_ and 0.1 mg/L CaCl_2_) containing Noble agar (1.0%) in the presence and absence of Ga at varying concentrations. The concentration of Ga that visibly decreased colony formation in the *recA* mutant and produced no growth changes in the negative control strains was selected as the sublethal inhibitory concentration. Furthermore, Δ*recA*, Δ*lacA*, and Δ*lacY* and the WT strain were grown overnight in the presence of ionic nitrate at the equivalent molarity as the sublethal inhibitory concentration to ensure growth was not influenced by the accompanying counter ion. In order to identify Ga-sensitive and -resistant genes in this study, the Keio collection was exposed to 100 μM Ga(NO_3_)_3_ (Ga). Gallium nitrate was obtained from Sigma–Aldrich, St. Louis, MO, USA. Stock solutions of Ga were prepared with deionized H_2_O and stored in glass vials for no longer than two weeks.

Similarly, Δ*recA*, Δ*lacA*, and Δ*lacY* and the WT strain were grown on M9 minimal media plates in the presence of varying concentrations of hydroxyurea (HU), obtained from USBiological Salmen, MA, USA, or sulfometuron methyl (SMM) obtained from Chem Service, West Chester, PA, USA, dissolved in ddH_2_O and dimethyl sulfoxide, respectively. Select mutants from the Keio collection were exposed to a final concentration of 5.0 mg/mL HU and 5.0 μg/mL SMM in the presence and absence of 100 μM Ga(NO_3_)_3_.

### 2.3. Screening

M9 minimal media and Noble agar (1.0%) plates, with and without the addition of Ga, were prepared two days prior to use. Here, Ga was added directly to the liquid agar and swirled before solidification. Colony arrays in 96-format were produced and processed using a BM3 robot and spImager (S&P Robotics Inc., Toronto, ON, Canada), respectively. Cells were transferred from the arrayed microtiter plates using a 96-pin replicator onto Luria-Bertani (LB) media agar plates and grown overnight at 37 °C. Colonies were then transferred using the replicator onto two sets of M9 minimal media Noble agar plates, with and without 100 μM Ga(NO_3_)_3_. Plates were then grown overnight at 37 °C. All images were acquired using the spImager and colony size, a measure of Ga sensitivity or resistance, was determined using integrated image processing software. Three biological trials were conducted and each of these trials included four technical replicates originating from the 96-colony array, which were combined and expanded onto a single plate in 384-colony array format; *n* (trials) ≥ 9. Strains presenting less than nine replicates were excluded (see Section 2.5).

Select mutants were exposed to HU or SMM at sublethal inhibitory concentrations. Identical conditions were maintained to enable direct comparisons between mutants grown in the presence of Ga only, and those grown in the presence of Ga and either HU or SMM. Here HU or SMM were added to the M9 minimal media plates directly before solidification.

### 2.4. Normalization

In this study, incubation time and temperature, nutrient availability, colony location, agar plate imperfections, batch effects, and neighboring mutant fitness were considered independent variables that could influence colony size and subsequently cause systematic variation. As a result, the colonies were normalized and scored using Synthetic Genetic Array Tools 1.0 (SGATools) [37,38], a tool that associates mutant colony size with fitness, thereby enabling quantitative comparisons. All the plates were normalized to establish average colony size, working on the assumption that the majority of the colonies would exhibit WT fitness since the concentration of Ga used in this study was below the minimal inhibitory concentration.

Mutant colony sizes in the presence (challenge) and absence (control) of Ga were quantified, scored, and compared as deviation from the expected fitness of the WT strain. This assumes a multiplicative model and not an additive effect originating from the challenge. Once scored, mutants displaying a reduction in colony size were indicative of a Ga sensitive hit and those displaying an increase in colony size were recovered as Ga resistant hits. Finally, the *p*-value was calculated as a two-tailed *t*-test and significance was determined using the Benjamini–Hochberg procedure, as a method of lowering the false discovery rate, which was selected to be 10%.

### 2.5. Data Mining and Analyses

Data mining was performed using Pathway Tools Omics Dashboard, which surveys against the EcoCyc database [39] and Uniport [40]. This allowed for the clustering of the Ga resistant and sensitive data sets into systems, subsystems, and individual objects (Table A1). Here, genes can be found in multiple systems since many are involved in a number of cellular processes.

Enrichment analyses were performed using the DAVID Bioinformatics Resource 6.8 [41,42]. Moreover, as a means of revealing the direct (physical) and indirect (functional) protein interactions amongst the gene hits, the STRING database [43] was utilized. Node maps based on experimental, co-expression, and gene fusion studies were generated using the Ga resistant and sensitive hits found in our screen.

## 3. Results and Discussion

### 3.1. Genome-Wide Screen of Ga Resistant and Sensitive Hits

In this work, the chemical genetic screen provided a method for the identification of the non-essential genes that may be involved in Ga resistance or sensitivity. A total of 3985 non-essential genes were screened for growth in the presence of 100 μM Ga(NO_3_)_3_ and from here, 3641 hits, in which *n* ≥ 9, were used for subsequent statistical analyses (Figure 1 and Appendix A). The statistical cutoff that suggested a significant difference in fitness when compared to the WT, indicated by a change in colony size, was selected to be two standard deviations from the mean or a normalized score of +0.162 and −0.154. This resulted in 107 gene hits, which represents approximately 2.5% of the open reading frames in the *E. coli* K-12 genome. In general, the normalization was performed with the assumption that hits presenting scores within two standard deviations from the mean had non-specific or neutral interactions with Ga. Therefore, the remaining hits were not regarded as significant based exclusively on the cutoffs selected.

In this work, the absence of the gene was inferred to give rise to the Ga resistant or -sensitive phenotype. A decrease in colony size (normalized score < −0.154) signified a Ga-sensitive hit, which implied that the presence of this gene increased Ga resistance. Here, 58 genes were found to cause Ga sensitivity when absent (Table 1). Likewise, an increase in colony size (normalized score > 0.162) signified a Ga-resistant hit, therefore the presence of this gene may suggest an increase in toxicity. Comparably, 49 genes were found to impart resistance when absent (Table 2), within the cutoffs applied.

Using Pathway Tools, which surveys against the EcoCyc database, a number of gene hits were mapped to more than one system and subsystem (Table 1 and Table 2). In general, comparable number of hits were mapped to the system “Response to stimulus”, “Cellular processes”, “Energy”, and “Biosynthesis” (Figure 2). Still, “Regulation”, “Degradation”, and proteins of the “Cell exterior” contained more resistant hits. Whereas “Other pathways” and proteins involved in processes of the “Central dogma” were represented by the Ga-sensitive hits at least two-fold more than the Ga resistant hits (Figure 2). Proteins residing or involved in maintaining cell envelope homeostasis were not enriched in the resistant hits; however, two-fold more hits were mapped to the system “Cell exterior” using EcoCyc’s system of classification when compared to the sensitive hits (Figure 2).

Despite similar numbers of resistant and sensitive hits scored in this screen, a greater number of categories were enriched for by the resistant hits, such as the biosynthesis of the vital coenzyme— biotin—when surveyed using the DAVID gene functional classification (Figure 3).

In addition, a number of amino acid biosynthetic processes and cytosolic proteins were enriched in the resistant hits, whereas proteins involved in the processing of 20S pre-rRNA and malate metabolic processes were enriched in the sensitive hits (Figure 3). In general, the enrichment profile of the resistant and sensitive hits provides insight into the dissimilarities between the mechanisms of Ga toxicity and resistance since there was no overlap in enrichment (Figure 3). Based on previous reports [44,45] several mutants belonging to the Keio collection, such as those involved in the synthesis of amino acids, did not grow in M9 minimal media, contrary to what we observed in this study. We attribute this observation to the presence of residual resources, such as amino acids, that were carried over from the LB media agar plates onto to the M9 minimal media agar plates. Once these resources are exhausted, dying cells may provide a source of nutrients for surviving cells. Furthermore, previous studies have provided cutoff values as markers of growth, such as one-third the average OD [45]. Mutants displaying growth below the cutoff are regarded as non-growers despite possible survival. As a follow up, we grew a number of mutants overnight, including *leuC*, *metA*, *proA, ilvB*, *trpD*, *lacA,* and the WT strain in liquid M9 minimal media from existing culture stocks (see Section 2.1) and transferred 20 μL onto M9 minimal agar plates in the absence and presence of Ga. These strains were then grown overnight. Growth was only observed for the WT strain and the *lacA* mutant. When the same procedure was completed with liquid LB medium and agar plates, colony formation was evident for each mutant tested. As a result, in this study we were able to test mutants that have otherwise been reported to not grow on minimal media due to the lack of essential nutrients, such as amino acids.

### 3.2. Ga Sensitive Systems

#### 3.2.1. Iron Homeostasis and Transport, and Fe–Sulfur Cluster Proteins

Gallium(III) has been shown to disrupt the function of several enzymes containing Fe–sulfur clusters, likely by competing for Fe-binding sites [7]. *Escherichia coli* contains over 10 Fe-acquisition systems, encoded by over 35 genes [46], providing an abundance of Ga potential targets, such as the sensitive hit *fdx* (ferredoxin). The protein product of *fdx* serves as an electron transfer protein in a wide variety of metabolic reactions, including the assembly of Fe–sulfur clusters [47], consequently, Ga resistance is probable if this metal is damaging Fe–sulfur centers. Ferredoxin may also serve as a binding site since the exchange of Fe may cause Ga sequestration. Furthermore, the sensitive hit *lipA* (lipoyl synthase) codes for an enzyme that uses ferredoxin as a reducing source, and catalytic Fe–sulfur clusters to produce lipoate [48]. This protein’s requirement for ferredoxin may provide an explanation for the two-fold score decrease observed in the *lipA* mutant when compared to the *fdx* mutant. Furthermore, our screen recovered the hit *ygfZ*, which codes for a folate-binding protein that is implicated in protein assembly and the repair of Fe–sulfur clusters [49]. The loss of *ygfZ* results in sensitivity to oxidative stress likely due to the generation of ROS, subsequently, this may lead to the inhibition of Fe–sulfur cluster assembly or repair [50]. In addition, the disruption of Fe–sulfur clusters has been found to downregulate the uridine thiolation of particular tRNAs as a means of decreasing sulfur consumption [51]. This process appears to be important in coupling translation with levels of sulfur-containing amino acids. We recovered *trmU*, which encodes a tRNA thiouridylase as a Ga-sensitive hit in this study.

The redox pair Fe(II)/Fe(III) is well suited for a number of redox reactions and electron transfers. Accordingly, bacteria have developed a number of Fe-acquisition systems, such as siderophores and Fe-chelating proteins [52]. Siderophores, such as enterobactin are synthesized internally and exported extracellularly to scavenge Fe(III) from the environment [53]. The ferric-siderophore complex is imported into the cell and then degraded to release Fe(III) [53] and since Ga is an Fe mimetic [54], this metal has been demonstrated to bind certain siderophores [23]. The protein TolC is an outer membrane carrier required for the export of the high-affinity siderophore enterobactin from the periplasm to the external environment [55]. The Ga sensitivity of the Δ*tolC* strain may be due to the periplasmic accumulation of Ga-enterobactin complexes. If TolC is inactivated, then less enterobactin is exported outside the cell in turn providing more Ga targets, and as a result, Ga-enterobactin complexes may accumulate inside the cell. Further, EvgA is part of the EvgAS two-component system involved in the transcriptional regulation of *tolC* [56]. Loss of *evgA* is expected to display a similar defect in enterobactin export as would a *tolC* mutant, thus resulting in Ga sensitivity. Finally, *bfr* (bacterioferritin) was recovered as a sensitive hit in this work. This protein, which binds one heme group per dimer and two Fe atoms per subunit, functions in Fe storage and oxidation [57]. The sensitivity phenotype of the Δ*bfr* strain may be associated with a failure to mitigate Fe-mediated ROS production due to the disruption of Fe homeostasis in the presence of Ga (see Section 3.2.2).

#### 3.2.2. Oxidative Stress

The production of ROS has been shown to be a mechanism of metal toxicity. Exposure to hydrogen peroxide or other agents that catalyze the production of ROS, such as superoxide, causes DNA and protein damage to macromolecules including proteins, lipids, nucleic acids and carbohydrates [58]. This in turn causes the upregulation of genes encoding ROS-scavenging enzymes [58]. An increase in cytoplasmic Fe intensifies ROS toxicity by catalyzing the exchange of electrons from donor to hydrogen peroxide [22]. Consequently, this may require the assistance of cellular antioxidants such as glutathione, and enzymes such as catalase, superoxide dismutase and peroxidase [59]. Gallium(III) is Fenton inactive, and therefore the induction of ROS in the presence of Ga is likely to result in the release of Fe in the cytoplasm. One study observed higher levels of oxidized lipids and proteins in Ga exposed *Pseudomonas fluorescens* [7]. In turn, the oxidative environment stimulated the synthesis of nicotinamide adenine dinucleotide phosphate (NADPH) via the overexpression of NADPH-producing enzymes, invoking a reductive environment.

In this screen, several sensitive Ga hits effective in ROS protection were recovered, including γ-Glutamate-cysteine ligase, or *gshA*. Strains lacking this gene have been shown to be hypersensitive to thiol-specific damage generated through mercury and arsenite exposure [60]. Similarly, strains lacking glyoxalase II (*gloB*), also a sensitive hit in this study, accumulate *S*-lactoylglutathione and demonstrate depleted glutathione pools [61]. If this antioxidant is depleted, then the potential for ROS-mediated protection is lowered. Furthermore, the gene *grxD*, which codes for a scaffold protein that transfers intact Fe–sulfur clusters to ferredoxin, was also recovered as a Ga-sensitive hit. The presence of this abundant protein is further upregulated during stationary phase [62] and one study demonstrated, using the Keio collection, that a *grxD* mutant is sensitive to Fe depletion [63]. Based on this observation, Ga exposure may prompt toxicity via Fe exhaustion, or the introduction of this toxin may result in ROS production thereby leading to Fe–sulfur damage. Finally, bacterioferritin (*bfr*) was also identified as a sensitive hit in this work. This protein acts to prevent the formation of hydrogen peroxide from the oxidation of Fe(II) atoms [57]. The sensitivity phenotypes of the Δ*gshA*, Δ*gloB*, Δ*grxD,* and Δ*bfr* strains may be associated with Fe-mediated ROS production upon the disruption of Fe-homeostasis in Ga exposed cells.

The sensitive hit *ubiG,* involved in the production of ubiquinol-*8*, a key electron carrier used in the presence of oxygen or nitrogen, was recovered in this screen. The production of ubiquinol from 4-hydroxybenzoate and trans-octaprenyl diphosphate necessitates the use of six enzymes and UbiG twice [64]. Mutant strains deficient in ubiquinol demonstrate higher levels of ROS in the cytoplasmic membranes, a threat lessened via the addition of exogenous ubiquinol [65]. Furthermore, the Δ*ubiG* strain exhibited reduced fitness when exposed to oxidative stress [65]. Altogether, the presence of this hit may be explained by the exacerbation of the production of ROS due to Ga exposure alongside the compromised oxidative stress response of the Δ*ubiG* strain.

#### 3.2.3. Deoxynucleotide and Cofactor Biosynthesis, and DNA Replication and Repair

Compounds targeting ribonucleotide reductase (RNR), a key enzyme involved in the synthesis of deoxynucleotides from ribonucleotides, have long been regarded as cancer therapeutics [66]. In mammalian cells, Ga targets RNR through at least two mechanisms. These mechanisms include the inhibition of cellular Fe uptake resulting in decreased Fe availability at the M2 subunit of the enzyme [67] and direct inhibition of RNR activity [68], leading to a reduction in the concentration of nucleotides in the cell. This mechanism is not limited to mammalian cells. Gallium(III) has been shown to inhibit RNR and aconitase activity in *M. tuberculosis* [16]. If RNR inhibition is in fact a mechanism of Ga toxicity, then we predict that gene deletions resulting in decreased deoxynucleotide levels may cause hypersensitivity. Consequently, the deletion of the gene *purT*, which is involved in purine nucleotide biosynthesis [69], resulted in Ga sensitivity in this study.

Chromosomal replication is delayed in *E. coli* cells when the deoxynucleotide pool is depleted upon the inhibition of RNR [70]. If this is the case, then a defect in DNA replication may result in hypersensitivity to Ga. Our observation that the loss of the DNA polymerase III subunits HolC and HolD causes Ga sensitivity appears to support this hypothesis. Another potential consequence of RNR inhibition is an increase in stalled replication forks, which are prone to DNA strand breakage [70]. Resumption of stalled replication forks and double strand breaks due to defective RNR function require the activity of recombination repair enzymes such as the RuvABC, RecBCD and RecA [71,72]. Our results support these observations since the deletion of *recA*, *recD* or *ruvC* triggered the Ga sensitive phenotype. It is important to note that genes involved in base and nucleotide excision repair were not retrieved as Ga sensitive hits suggesting that DNA damage associated with Ga exposure may be predominantly in the form double stranded breaks.

A number of sensitive hits were mapped to the subsystem “Biosynthesis of cofactors, prosthetic groups and electron carriers”. Processes affected include folate, lipoate, quinol, quinone, ubiquinol and thiamine biosynthesis. The gene products of *pabA* and *pabC*, which encode an aminodeoxychorismate synthase and an aminodeoxychorismate lyase, respectively, are involved in the biosynthesis of p-aminobenzoic acid [73], a precursor of folate. In both prokaryotes and eukaryotes, folate cofactors are necessary for a range of biosynthetic processes including purine and methionine biosynthesis (Figure 4) [74]. Folate biosynthesis has long served as an antibiotic target in prokaryotes since this cofactor is synthesized only in bacteria yet actively imported by eukaryotes using membrane associated processes [75]. Similar to *purT*, the Ga sensitivity of Δ*pabA* and Δ*pabC* strains may be a result of the reduction in deoxynucleotide levels caused by the inactivation of RNR.

To test the potential connection between Ga and RNR activity, we exposed the *holC*, *holD*, *recA*, *recD*, *ruvC* and *purT* mutants to hydroxyurea (HU), which is a known inhibitor of RNR activity [76]. Further, we included a number of mutants involved in DNA synthesis, such as *ruvA* and *recR*, that were not uncovered in our initial screen. In *E. coli*, HU has been shown to increase ribonucleotide pools and decrease total deoxyribonucleotide concentrations, thus negatively affecting the synthesis of DNA [77]. We exposed these mutants to sublethal concentrations of HU and normalized the cellular effect of this agent. Using this reagent, the sensitivity of the *holC*, *ruvC* and *recD* mutants in the presence of HU and Ga was found to increase (Table 3). Furthermore, *ruvA*, which assists in recombinational repair together with *ruvB* [78], was also found to be a sensitive hit in the presence of this inhibitor. The genes *purT* and *holD* were not uncovered as either sensitive or resistant hits based on the cutoffs applied and no changes in the sensitivity or resistance of either *lacA* or *lacY*, negative controls in this work, were statistically identified.

### 3.3. Systems Involved in Ga Resistance

#### 3.3.1. Fe Transport Systems

In *E. coli*, the mechanisms by which Ga is transported into the cell have yet to be identified. In this screen, we identified a number of transport proteins that confer resistance against Ga when absent. Metal resistance mechanisms may involve decreased import or enhanced export of the toxin. Therefore, loss of a gene in which the product mediates import of the toxin into the cell would prevent its accumulation and result in resistance. Both FepG and TonB are proteins that demonstrate close interaction (Figure 5) and fit the latter criterion, both involved in the import of Fe-siderophores. The protein FepG is an inner membrane subunit of the ferric enterobactin ATP-binding cassette transporter complex. When *fepG* is inactivated, *E. coli* cells lose ferric enterobactin uptake abilities [79,80]. The protein product of *tonB* is a component of the Ton system which functions to couple energy from the proton motive force with the active transport of Fe-siderophore complexes and Vitamin B12 across the outer membrane [81]. Since Ga entry into the bacterial cell can occur through siderophore binding and since this metal is an Fe mimetic [23,54], we hypothesize that in the absence of *fepG* and *tonB* Ga import and intracellular accumulation is reduced.

OmpC is a promiscuous porin that permits the transport of 30+ molecules, and is postulated to be a transporter of copper(I) and copper(II) [82] and potentially other metal species [83]. It has been hypothesized that Ga can cross the membrane of *E. coli* via porins [23]. While this hypothesis has not been demonstrated in *E. coli* directly, other works have confirmed findings in *P. aeruginosa* [9], *Mycobacterium smegmatis* [84] and *Francisella* strains [12]. Further evidence for the importance of OmpC in Ga resistance can be visualized using the STRING map (Figure 5). Here, OmpC is connected to two proteins that comprise the ATPase complex through the periplasmic protein TolB. TolB has been shown to physically interact with porins such as OmpC and is required for their assembly into the outer membrane of *E. coli* cells [85]. The resistance recovered in the Δ*tolB* strain may be due to a disruption in OmpC function, thereby hindering Ga import. In addition, CysU, which is involved in the uptake of sulfate and thiosulfate was also recovered as a resistant hit [86]. According to the hard-soft acid-base theory, Ga coordinates well with sulfate or thiosulfate [4]. A reduction in the uptake of these metabolites may prove useful against Ga stress due to decreased toxin import.

Genes involved in Fe import in other organisms have been shown to confer Ga resistance when deleted, or Ga sensitivity when overexpressed. A three-fold increase in Ga resistance was displayed upon the deletion of the gene *hitA*, which codes for a Fe-binding protein in *P. aeruginosa* [25]. The *Haemophilus influenzae* proteins FbpABC, which are involved in the delivery of Fe from the periplasm to the cytoplasm, were expressed in *E. coli* as a means of investigating their impact on Ga import, which increased in the presence of these genes [87]. Furthermore, earlier studies have examined the use of metal-chelators as antimicrobial enhancements. Although the majority of studies regarding Ga import have been performed in *P. aeruginosa*, some findings can be compared. For example, it has been demonstrated that the siderophore complex Ga-deferoxamine was slightly more effective at killing cells than Ga alone [10] and more promising results have been made with the complex Ga-protoporphyrin IX [88]. Altogether, these studies and our work suggest that Ga enters the cell via siderophore transport systems or Fe-binding transporters.

#### 3.3.2. Amino Acid Biosynthesis

Ga resistant hits were functionally enriched for the synthesis of amino acids (Figure 3), classified in the subsystem, “Amino acid biosynthesis” (Table 2) and highly connected in the functional map (Figure 5). The genes recovered were found to be mainly involved in the biosynthesis of branched (*ilvB*, *ilvY*, *leuA* and *leuC*) and aromatic (*aroF*, *trpB*, and *trpD*) amino acids, methionine (*metA* and *metR*), and proline (*proA* and *proB*). The demand for NADPH in biosynthetic pathways of branched and aromatic amino acids, as well as methionine and proline, are among the highest [89]. It is plausible that a defect in the synthesis of these amino acids may increase levels of NADPH, which has been shown to neutralize the oxidative stress elicited from Ga exposure [7].

To further test this hypothesis, we exposed a number of the resistant hits mapped to branched amino acid biosynthesis to sublethal concentrations of Sulfometuron methyl (SMM), an inhibitor of acetolactate synthase [90], a key enzyme involved in the synthesis of branched amino acids. The resistance score of *ilvY* and *leuA* increased in the presence of SMM (Table 4). Sulfometuron methyl inhibits acetolactate synthase, which in turn may increase the liable NADPH pool. In fact, *ilvY* is a positive regulator of *ilvC* [91], which encodes a reductoisomerase and is the only enzyme in this pathway that directly uses NADPH. Here, *ilvB* and other genes involved in branched amino acid biosynthesis did not make the statistical cutoffs owing to large standard deviations. Finally, no changes in the sensitivity or resistance of *lacA* or *lacY*, negative controls in this work, were statistically identified.

It has been postulated that the oxidation of amino acids is a common and damaging effect of metal-induced oxidative stress [92]. Certain side chains, such as Arg, Cys, His, Lys and Pro residues are major targets, leading to protein damage and intra/inter-crosslinking [92,93]. If Ga targets amino acids, both free and within proteins, a possible explanation for the recovery of amino acid gene resistant hits in this study may rest in the cell’s requirement to repair or replace damaged amino acids. If these genes are absent fewer Ga targets remain and the cell expends less energy rebuilding these targeted biomolecules, while directing more energy elsewhere, such as scavenging and importing required metabolites. Furthermore, the oxidation of these amino acid side chains may lead to the propagation of ROS, and therefore a deficiency in amino acids may minimize damage by slowing the advancement of amino acid metal-induced oxidative stress.

#### 3.3.3. Lipopolysaccharides and Peptidoglycan

The *E. coli* envelope is composed of lipopolysaccharides (LPS), which surround and protect the cytoplasm, and the cross-linked polymer peptidoglycan (PG), which is the primary stress-bearing biomolecule in the cell [94]. In this study, a number of genes involved in LPS or PG biosynthesis/maintenance were observed to cause Ga resistance when absent. These genes include *cpsG* and *rfaC* (LPS), and *alr*, *env* and *mrcB* (PG). Many of these genes are RpoS-regulated and participate in maintaining membrane integrity in response to pressure [95]. Loss of *mrcB*, which encodes for an inner membrane enzyme functioning in transglycosylation and transpeptidation of PG, has been shown to result in reduced surface PG density when absent [96]. The protein RfaC is essential in LPS production [97] and cells lacking this gene contain defects in the core heptose region [98]. The protein EnvC, which is a divisome-associated factor has been shown to have PG hydrolytic activity and result in decreased cell envelope integrity when deleted. Furthermore, the protein product of *tolB*, which plays a role in maintaining the structure of the cell envelope, was also a Ga-resistant hit. Cells deficient in *tolB* have been shown to release periplasmic proteins into the extracellular space [99]. An explanation for the appearance of *mrcB*, *envC* and *tolB* in this study may reside in the ability of PG to bind metals. Metal ions are known to bind the LPS or PG layer of Gram-negative and Gram-positive bacteria [100], and the presence of anionic groups such as carboxylic acids [101] and other hard acids within the cell envelope, provide suitable binding sights for free metal ions like Ga. Although the major ionic form of Ga is Ga(OH)_4_^−^, free Ga ions produced through equilibrium may be quickly bound by hard acids such as alcohols, carboxylates, and hydroxyls, which comprise the bulk of the PG. Despite their presence at low concentrations these species may further impede cell health and cause toxicity. However, if the LPS or PG layer is reduced, as would be the case in the absence of *mrcB*, *rfaC*, *envC* and *tolB*, then a reduction in Ga-cell envelope binding may occur. In the case of the Δ*tolB* strain, the potential release of periplasmic proteins with Ga-binding sites into the extracellular space may also provide protection via sequestration, which is a common bacterial resistance mechanism [24]. Another possible explanation for Ga resistance associated with LPS and PG genes may include the structural alteration of the cell envelope, which may disrupt Fe import systems. Inhibition of lipid biosynthesis prevents proper assembly and insertion of porins into the outer membrane since LPS-porin interaction sites have been shown to be important in their biogenesis [102,103]. Therefore, compromised function of siderophore receptors or porins in these mutants could decrease Ga import and mitigate toxicity.

## 4. Conclusions

In this study, the Keio collection was used as a means of drawing insight into the mechanisms of Ga toxicity and resistance in *E. coli* BW25113. In total, 3895 non-essential genes were screened and 3641 of these were normalized and scored. Genes demonstrating resistance or toxicity were mined to highlight processes and pathways affected by Ga exposure. Mutants demonstrating an increase in colony formation were considered resistant hits, in that the presence of the gene results in Ga sensitivity. In contrast, a decrease in colony size was regarded as a Ga-sensitive hit, consequently it was assumed that the presence of this gene would impart the resistant phenotype and mitigate the toxicity of prolonged Ga exposure.

Overall, comparable numbers of resistant and sensitive hits were mapped to each subsystem using Pathway Tools, which surveys against the EcoCyc Database. When examining the fold enrichment data, no biological process was enriched comparably between the two data sets. One general observation made evident from the latter conclusion is that distinct pathways are affected by Ga when comparing the mechanisms of toxicity and resistance since no overlap in functional enrichment was uncovered. Still, one significant exception was found: Fe-metabolism. Based on this study, and previous reports, there is a relationship between Ga and Fe-metabolism. The genes that code for TonB and FepG were two resistant hits highlighted in this work. On the contrary, Fdx, Bfr and LipA, proteins also involved in Fe-metabolism, gave rise to sensitivity when absent. Therefore, we propose that Fe-metabolism may serve as a mechanism of resistance and toxicity in *E. coli*. Here, the complexity of Ga exposure is made further apparent, fostering more questions regarding the interaction of this metal with microbes. What is clear however, is that the mechanism of Ga action is likely a result of a number of direct and indirect interactions, an observation made evident by the wide array of hits uncovered in this work.

Few studies have explored the mechanisms of adaptive resistance in *E. coli* under sub-lethal concentrations of Ga. In response, we have presented a number of genes that are implicated to be involved in adaptive survival. For example, genes involved in preventing oxidative damage and DNA repair were emphasized as sensitive hits, as such that their presence gives rise to resistance. In short, preventing and repairing DNA damage, a mechanism that has yet to be demonstrated in vivo, and redox maintenance may provide tools by which microbial organisms mitigate metal stress.

The use of Ga for the treatment of diseases and infections is gaining considerable attention. Still, to further the development of this metal as an antimicrobial agent it is imperative that we determine the associated mechanisms of toxicity and resistance. Further work must be completed to specifically test the various hypotheses we have presented here, such as determining the mode of Ga entry, the levels of ROS produced in the cell and the specific influence of Ga on Fe-metabolism. Nonetheless, this study provides a significant number of biomolecular mechanistic hypotheses to the community investigating the mechanisms of Ga action in *E. coli* and other microbes.

## Figures and Tables

**Figure 1 genes-10-00034-f001:**
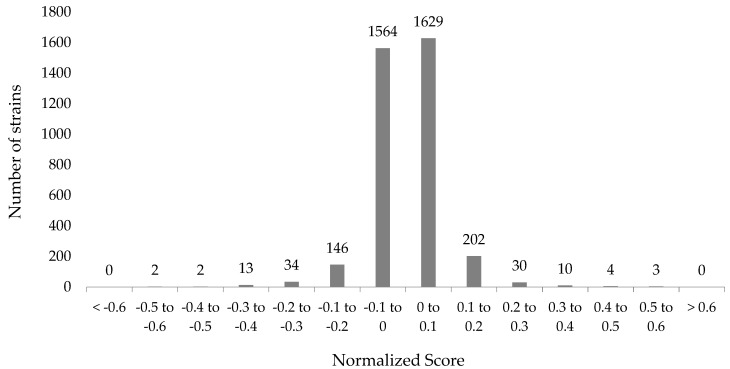
Synthetic Array Tools (version 1.0) was used to normalize and score the Gallium(III) (Ga) resistant and sensitive hits as a means of representing the growth differences in *Escherichia coli* K12 BW25113 in the presence of 100 μM Ga(NO_3_)_3_. Each individual score represents the mean of 9–12 trials.

**Figure 2 genes-10-00034-f002:**
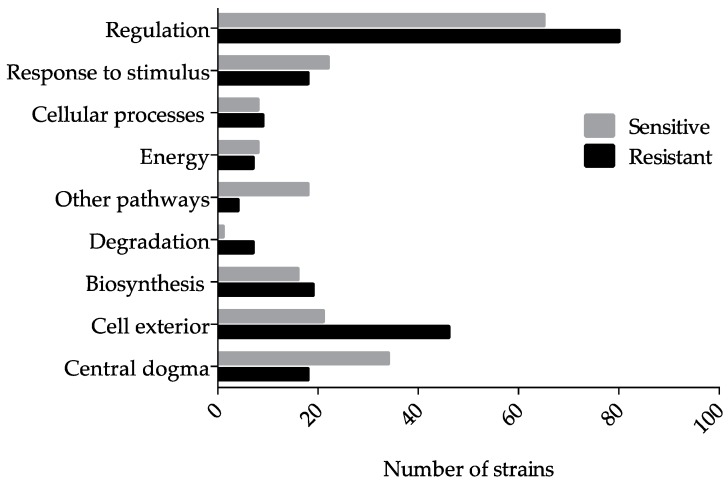
Ga-resistant and -sensitive gene hits mapped to component cellular processes. Several gene hits are mapped to more than one subsystem. The cutoff fitness score selected was two standard deviations from the mean and recovered gene hits with a score outside this range were chosen for further analyses. The hits were mined using the Omics Dashboard (Pathway Tools), which surveys against the EcoCyc database. Each individual score represents the mean of 9–12 trials.

**Figure 3 genes-10-00034-f003:**
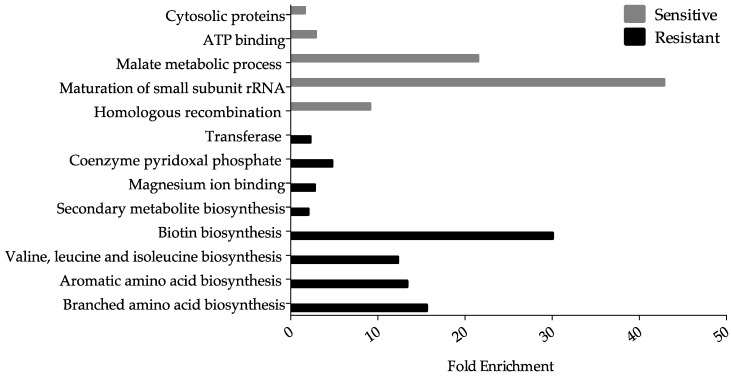
Functional enrichment among the Ga-resistant and -sensitive gene hits. The DAVID gene functional classification (version 6.8) database, a false discovery rate of 10% and a cutoff score two standard deviations from the mean was used to measure the magnitude of enrichment of the selected gene hits against the genome of *E. coli* K-12. Only processes with gene hits ≥3 were included.

**Figure 4 genes-10-00034-f004:**
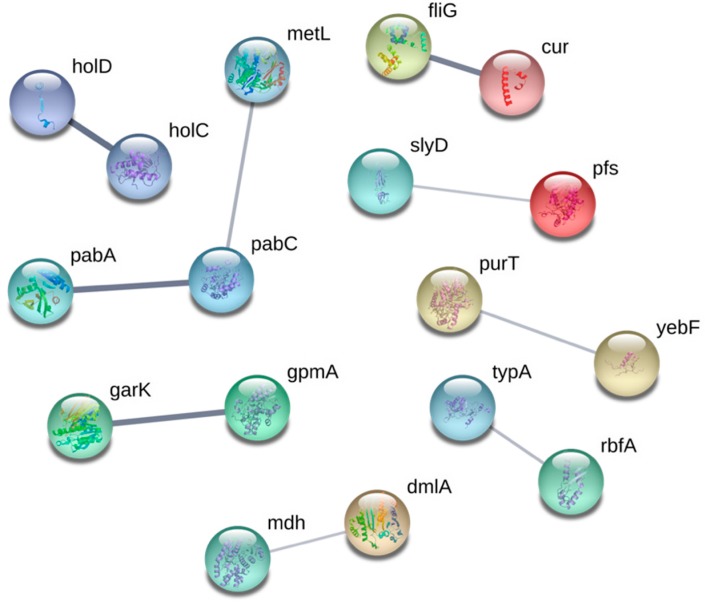
Connectivity map displaying the predicted functional associations between the Ga-sensitive gene hits; disconnected gene hits not shown. The thicknesses of the lines indicate the degree of confidence prediction for the given interaction, based on fusion, curated databases, experimental and co-expression evidence. Figure generated using STRING (version 10.5) and a medium confidence score of 0.4.

**Figure 5 genes-10-00034-f005:**
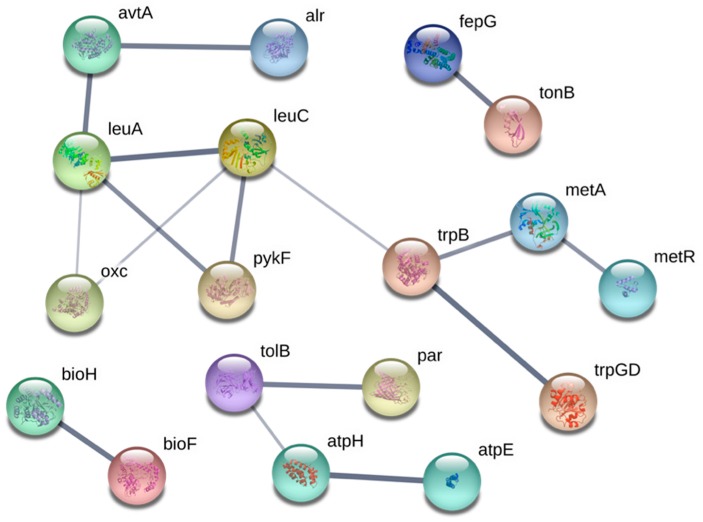
Connectivity map displaying the predicted functional associations between the Ga-resistant gene hits; disconnected gene hits not shown. The thicknesses of the lines indicate the degree of confidence prediction for the given interaction, based on fusion, curated database, experimental and co-expression evidence. Figure generated using STRING (version 10.5) and a medium confidence score of 0.4.

**Table 1 genes-10-00034-t001:** Ga sensitive hits organized according to system and subsystem mined using the Omics Dashboard (Pathway Tools), which surveys against the EcoCyc Database; genes represent sensitive hits with scores < −0.154.

System	Subsystem	Gene ^1^	Score ^2,3^
Central dogma	Transcription	*evgA*	−0.166
		*hns*	−0.175
		*lgoR*	−0.401
		*nagC*	−0.191
		*rseA*	−0.26
		*ulaR*	−0.556
	Translation	*bipA*	−0.204
	DNA metabolism	*holC*	−0.327
		*holD*	−0.217
		*ruvC*	−0.184
		*intR*	−0.27
		*recA*	−0.309
		*recD*	−0.199
	RNA metabolism	*rbfA*	−0.35
		*rim*	−0.298
		*mnmA*	−0.212
		*rnt*	−0.322
		*ygfZ*	−0.373
		*evgA*	−0.166
		*hns*	−0.175
		*lgoR*	−0.401
		*nagC*	−0.191
		*rseA*	−0.269
		*sspA*	−0.214
		*ulaR*	−0.556
	Protein metabolism	*lipA*	−0.318
		*pphA*	−0.198
		*slyD*	−0.273
	Protein folding and secretion	*slyD*	−0.273
Cell exterior	Transport	*zunC*	−0.361
		*tolC*	−0.539
		*ugpC*	−0.29
	Pilus	*ybgO*	−0.163
	Flagellum	*fliG*	−0.235
	Outer membrane	*tolC*	−0.539
	Plasma membrane	*clsA*	−0.171
		*cysQ*	−0.203
		*fdnI*	−0.251
		*fliG*	−0.235
		*gspA*	−0.199
		*hokA*	−0.181
		*nuoK*	−0.247
		*rseA*	−0.269
		*ubiG*	−0.265
		*ugpC*	−0.29
		*znuC*	−0.361
	Periplasm	*tolC*	−0.539
		*yebF*	−0.268
Biosynthesis	Amino acid	*dmI*	−0.418
		*metL*	−0.189
		*mtn*	−0.329
	Nucleoside and nucleotide	*purT*	−0.216
	Fatty acid/lipid	*clsA*	−0.171
	Carbohydrate	*mdh*	−0.287
	Secondary metabolites	*mtn*	−0.329
		*fdx*	−0.168
	Cofactor	*fdx*	−0.168
		*gshA*	−0.165
		*lipA*	−0.318
		*pabA*	−0.224
		*pabC*	−0.258
		*ubiG*	−0.265
	Other	*metL*	−0.189
Degradation	Amino acid	*astD*	−0.301
	Nucleoside and nucleotide	*mtn*	−0.329
	Amine	*purT*	−0.216
	Carbohydrate	*garK*	−0.173
		*dmlA*	−0.418
Energy	Glycolysis	*gpmA*	−0.175
	Tricarboxylic acid cycle	*mdh*	−0.287
	Fermentation	*mdh*	−0.287
	Aerobic respiration	*nuoK*	−0.247
	Anaerobic respiration	*fdnI*	−0.251
		*nuoK*	−0.247
	Other	*mdh*	−0.287
		*nuoK*	−0.247
Cellular processes	Biofilm	*hns*	−0.175
	Adhesion	*ybgO*	−0.163
	Locomotion	*fliG*	−0.235
		*recA*	−0.309
	Viral response	*intR*	−0.27
	Host interaction	*intR*	−0.27
		*slyD*	−0.273
	Symbiosis	*slyD*	−0.273
Response to stimulus	Starvation	*sspA*	−0.29
		*ugpC*	−0.214
	Heat	*bipA*	−0.204
		*gloB*	−0.297
		*slyD*	−0.273
	Cold	*bipA*	−0.204
		*rbfA*	−0.35
	DNA damage	*rbfA*	−0.35
		*recA*	−0.39
		*recD*	−0.199
		*ruvC*	−0.184
	Osmotic stress	*gshA*	−0.165
		*ubiG*	−0.265
	Other	*evgA*	−0.166
		*fliG*	−0.235
		*grxD*	−0.266
		*holC*	−0.327
		*holD*	−0.217
		*pphA*	−0.198
		*rseA*	−0.269
		*sspA*	−0.214
		*tolC*	−0.539
		*ugpC*	−0.29
Other pathways	Inorganic nutrient metabolism	*fdnI*	−0.251
		*nuoK*	−0.247
	Detoxification	*gloB*	−0.297
		*grxD*	−0.266
	Macromolecule modification	*mnmA*	−0.212
		*rnt*	−0.322
	Other enzymes	*bfr*	−0.17
		*cysQ*	−0.203
		*pphA*	−0.198
		*recD*	−0.199
		*ruvC*	−0.184
		*slyD*	−0.273

^1^ Gene hits can be mapped to more than one system and subsystem. ^2^ Each individual score represents the mean of 9–12 trials. ^3^ Two-tailed *t*-test and significance was determined using the Benjamini–Hochberg procedure; false discovery rate 10%.

**Table 2 genes-10-00034-t002:** Ga-resistant hits organized according to system and subsystem mined using the Omics Dashboard (Pathway Tools), which surveys against the EcoCyc Database; genes represent resistant hits with scores >0.162.

System	Subsystem	Gene ^1^	Score ^2,3^
Central dogma	Transcription	*ilvY*	0.215
		*metR*	0.372
		*odhR*	0.353
	DNA metabolism	*hofM*	0.62
		*xerD*	0.168
		*cas2*	0.177
	RNA metabolism	*symE*	0.177
		*ilvY*	0.215
		*metR*	0.372
		*pdhR*	0.353
	Protein metabolism	*mrcB*	0.249
	Protein folding and secretion	*yraI*	0.18
Cell exterior	Transport	*cysU*	0.362
		*fepG*	0.312
		*tonB*	0.341
		*caiT*	0.403
		*yiaO*	0.6
		*par*	0.266
	Cell wall biogenesis	*alr*	0.353
		*evnC*	0.203
		*mrcB*	0.249
		*yraI*	0.18
	Lipopolysaccharide metabolism	*cspG*	0.204
		*rfaC*	0.201
	Outer membrane	*par*	0.266
		*pqiC*	0.345
	Plasma membrane	*atpE*	0.172
		*atpH*	0.176
		*caiT*	0.403
		*cycU*	0.362
		*envU*	0.203
		*fepG*	0.312
		*mrcB*	0.249
		*pqiC*	0.345
		*tonB*	0.341
		*torC*	0.259
		*rfaC*	0.201
		*yaaU*	0.237
		*yafU*	0.214
		*yifK*	0.18
	Periplasm	*ansB*	0.204
		*asr*	0.247
		*envC*	0.203
		*mrcB*	0.249
		*pqiC*	0.345
		*tolB*	0.2
		*tonB*	0.341
		*torC*	0.259
		*yiaO*	0.6
		*yral*	0.18
	Cell wall component	*mrcB*	0.249
		*torC*	0.259
Biosynthesis	Amino acid	*alr*	0.353
		*avtA*	0.384
		*leuA*	0.302
		*leuC*	0.205
		*metA*	0.241
		*proB*	0.258
		*trpB*	0.611
		*trpD*	0.273
	Fatty acid/lipid	*rfaC*	0.201
	Carbohydrate	*cpsG*	0.204
		*rfaC*	0.201
	Cofactor, prosthetic groups, electron carrier	*bioF*	0.183
		*bioH*	0.194
		*coaA*	0.193
		*thiE*	0.226
	Cell structure	*mrcB*	0.249
	Other	*aroF*	0.236
Degradation	Amino acid	*alr*	0.353
		*ansB*	0.204
	Fatty acid/lipid	*atoA*	0.246
Energy	Glycolysis	*pykF*	0.169
	Fermentation	*pykF*	0.169
	Anaerobic respiration	*torC*	0.259
	Adenosine triphosphate biosynthesis	*atpE*	0.172
		*atpH*	0.176
	Other	*hydN*	0.249
Cellular processes	Cell cycle/division	*envC*	0.203
		*tolB*	0.2
		*xerD*	0.168
	Cell death	*envC*	0.203
	Adhesion	*tonB*	0.341
	Viral response	*cas2*	0.177
		*tonB*	0.341
	Symbiosis	*tonB*	0.341
Response to stimulus	Heat	*pykF*	0.169
	DNA damage	*par*	0.266
		*symE*	0.177
		*yiaO*	0.6
	pH	*oxc*	0.519
	Other	*asr*	0.247
		*caiT*	0.403
		*cas2*	0.177
		*envC*	0.203
		*mrcB*	0.249
		*tolB*	0.2
		*tonB*	0.341
		*torC*	0.259
		*xerD*	0.168
		*yaaU*	0.237
Other pathways	Other enzymes	*oxc*	0.519
		*sepG*	0.201

^1^ Gene hits can be mapped to more than one system and subsystem. ^2^ Each individual score represents the mean of 9–12 trials. ^3^ Two-tailed *t*-test and significance was determined using the Benjamini–Hochberg procedure; false discovery rate 10%.

**Table 3 genes-10-00034-t003:** Hydroxyurea sensitive and gene hits involved in the synthesis of DNA, normalized to include only the effects of Ga exposure; those with a score two deviations from the mean are included.

Gene	Score without HA	Score with HA ^1,2^
*ruvA*	N/A	−0.257
*recA*	−0.309	−0.299
*ruvC*	−0.184	−0.299
*holC*	−0.327	−0.351
*recD*	−0.199	−0.561

^1^ Each individual score represents the mean of 9–12 trials. ^2^ Two-tailed *t*-test and significance was determined using the Benjamini–Hochberg; procedure; false discovery rate 10%. HA: Hydroxyurea

**Table 4 genes-10-00034-t004:** Sulfometuron methyl resistant gene hits, involved in the synthesis of amino acids, normalized to include only the effects of Ga exposure; only those with a score two deviations from the mean are included.

Gene	Score without SMM	Score with SMM ^1,2^
*leuA*	0.302	0.341
*ilvY*	0.215	0.3

^1^ Each individual score represents the mean of 9–12 trials. ^2^ Two-tailed *t*-test and significance was determined using the Benjamini–Hochberg; procedure; false discovery rate 10%. SMM: sulfometuron methyl

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
