# Peer review of "Using a Chemical Genetic Screen to Enhance Our Understanding of the Antimicrobial Properties of Gallium against Escherichia coli"

_genes, 2019, doi:10.3390/genes10010034_

Round 1
Reviewer 1 Report
The authors present a screening of the E. coli Keio mutant collection to understand the antimicrobial properties of Ga. The approach used here is the same of that used in a paper already published on Genes by the same authors, where the silver toxicity was investigated. I think that many readers could be potentially interested in the abovementioned topic. Therefore, the area of this study is not without merit. Nevertheless, I feel that the reported data is not robust to increase our knowledge on this topic. The approach utilized could be used to hypothesize the genes involved Ga response, but more experiments are needed to deepen the molecular mechanisms in which these genes are involved. Phenotypes often cannot be analysed only by the function of a single gene simply because it is not clear how regulatory activity will affect the biology of the whole microorganism. Ultimately, I feel that, when using such large collection, it would be appropriate to verify the final results of mutants of particular interest by PCR analysis and/or sequencing.
Some specific comments:
The introduction section should be focused on bacterial and gallium interactions;
Materials methods section should be rewritten deleting the parts already described (several sentences are “cut and paste” from Gugala et al., 2018). Moreover, several sentences are not appropriate for material and methods section (e.g. lines 116-119) and should be deleted.
Lines 156-162, please clarify: Was reduction/increase in colony size calculated in respect to WT, or Challenged samples, or both? I believe that a resistant mutant should show an increased Ga-resistance respect to WT.
Results: was the WT Ga-MIC evaluated?
Tables 1 and 2, the description of genes should be added, moreover the footnotes “1,2” should be cited in the table and not in the table caption.
Figures 4 and 5, the means of these two figures should accurately described or deleted. Please put the name of the genes in the captions.
Author Response
The authors present a screening of the E. coli Keio mutant collection to understand the antimicrobial properties of Ga. The approach used here is the same of that used in a paper already published on Genes by the same authors, where the silver toxicity was investigated. I think that many readers could be potentially interested in the abovementioned topic. Therefore, the area of this study is not without merit. Nevertheless, I feel that the reported data is not robust to increase our knowledge on this topic. The approach utilized could be used to hypothesize the genes involved Ga response, but more experiments are needed to deepen the molecular mechanisms in which these genes are involved. Phenotypes often cannot be analysed only by the function of a single gene simply because it is not clear how regulatory activity will affect the biology of the whole microorganism. Ultimately, I feel that, when using such large collection, it would be appropriate to verify the final results of mutants of particular interest by PCR analysis and/or sequencing.
- We agree with the reviewer on the need for further investigation, and while this would be the ideal, we strongly believe that publishing this information as a means of sharing it with the scientific community at this time is of foremost importance, which then allows for a community effort to follow up on the findings.
- We have not made any hard conclusions in this work, what we have delivered is a number of hypothesis that are to be further investigated. Many of these hypotheses have not been considered by the scientific community studying Ga because they have not been made known. However, now that we have presented them, researchers are able to expand their degree of scope.
- However, we have added several additional experiments aimed at drawing further insight into the mechanisms of Ga toxicity and resistance to complement the chemical genomics data.
The introduction section should be focused on bacterial and gallium interaction
- Here we have added additional information; particularly mechanisms of metal (and specifically Ga) resistance and toxicity in bacteria
- General statements were made as a means of providing content into the use of Ga and its prevalence in medicine
Materials methods section should be rewritten deleting the parts already described (several sentences are “cut and paste” from Gugala et al., 2018). Moreover, several sentences are not appropriate for material and methods section (e.g. lines 116-119) and should be deleted.
- We have removed segments of the methods and added several new methodologies
Lines 156-162, please clarify: Was reduction/increase in colony size calculated in respect to WT, or Challenged samples, or both? I believe that a resistant mutant should show an increased Ga-resistance respect to WT
- We have clarified in a number of places, including line 182-183. Colony sizes are compared to the expected fitness of the WT strain in the presence of the challenge
Results: was the WT Ga-MIC evaluated?
- Yes, this has been evaluated in an attempt to determine the sublethal Ga concentration used in our study and reported elsewhere, however we do not feel it is necessary to include this in this study
Tables 1 and 2, the description of genes should be added, moreover the footnotes “1,2” should be cited in the table and not in the table caption
- Gene description can be found in the supplementary table and the footnotes have been corrected
Figures 4 and 5, the means of these two figures should accurately described or deleted. Please put the name of the genes in the captions
- More information on the genes can be found in the supplementary table
Reviewer 2 Report
The manuscript "Using a Chemical Genetic Screen to Enhance Our Understanding of the Antimicrobial Properties of Gallium against Escherichia coli" by Gugala et al., describes a comprehensive genetic screen to identify bacterial genes involved in the mechanism of action of the antimicrobial agent gallium. The authors identified genes involved in both intrinsic and adaptive resistance to gallium, which certainly adds to the previous literature in the field. Significantly, Fe metabolism is highlighted as a core cellular process that may be involved in gallium activity. These results may enable the design of improved approaches to truly translate gallium as an antimicrobial.
The results presented are mostly descriptive. Can the authors elaborate further (in Conclusion section) on what they think are the mechanisms of action, and what are next steps to follow to elucidate this in more detail?
Furthermore, authors should cite relevant literature: FEMS Microbiol LEtt. 2018 Oct 1;365(20); Antimicrob Agents Chemother. 2012 May;56(5):2696-704. doi: 10.1128/AAC.00064-12;Appl Environ Microbiol. 2011 Aug;77(15):5220-9.doi:10.1128/AEM.00648-11.
Author Response
The results presented are mostly descriptive. Can the authors elaborate further (in Conclusion section) on what they think are the mechanisms of action, and what are next steps to follow to elucidate this in more detail?
- We have included several general statements regarding the mechanisms of Ga toxicity and resistance in the conclusion section
- Next steps or follow up experiments have also been briefly mentioned
Furthermore, authors should cite relevant literature: FEMS Microbiol LEtt. 2018 Oct 1;365(20); Antimicrob Agents Chemother. 2012 May;56(5):2696-704. doi: 10.1128/AAC.00064-12;Appl Environ Microbiol. 2011 Aug;77(15):5220-9.doi:10.1128/AEM.00648-11.
- We have reviewed both papers, the first is timely and is connected to our work based on the comparative use of microarray data. This is a nice study on effects of antimicrobial peptides and falls into the idea of evaluating antimicrobials system biology. However, since we are concentrating on the use of metals in this work, we believe it is important to stay within this realm. Many studies have used large scale omics approaches, such as microarrays and transcriptomic profiling, however the use of metals as antimicrobials is what is of importance to us. Regardless we thank the reviewer for pointing out these additional studies and we have cited the second paper as we found it relevant.
Reviewer 3 Report
I have read the manuscript Gugala et al with interest. The authors report the results of a chemical genomics screen to identify genes in E. coli that influence susceptibility of the cell to gallium toxicity, and describe the potential role of the identified genes. The manuscript is generally well written and presented. My specific comments are listed below, most of which are minor.
My only major comment relates to the interpretation of the results for genes involved in the biosynthesis of deoxynucleotides and amino acids. My understanding is that the chemical genomics screen was performed on M9 minimal medium. This medium would lack exogenous deoxynucleotides and amino acids. As such, mutations in the biosynthesis of these compounds are expected to be lethal both in the presence and absence of gallium - i.e., these mutants also should not grow on the control plate. Could the authors clarify why the deletion of these genes appeared to have an effect specifically in the presence of gallium, and why there were non-essential on the control plate?
Line 72: Cepacia should be lower case.
Lines 214-242: I found this section to be difficult to follow, and I suggest the authors attempt to rewrite it to improve clarity. I think the confusion arose, at least in part, as the discussion appears to switch back and forth between two distinct functional analyses (Figure 2 [EcoCyc categories] and 3 [DAVID categories]). Clarifying which analyses they are referring to at a given time may help improve the flow of this section.
Line 269-270: I do not follow the logic of this sentence. If TolC exports enterobactin from the periplasm to the external environment, would it not be expected to export unbounded enterobactin (so that it can then scavenge for iron). Why do the authors expect that a TolC mutant would therefore result in the accumulation of Ga-bounded enterobactin in the periplasm?
Line 293: gloB should be in italics.
Lines 454-456: This comment is related to my comment about Lines 214-242. The first sentence describes functional mapping with the EcoCyc categories, while the second sentence switches to describing results of the annotation with DAVID categories. However, the authors do not specify that these two sentences refer to distinct functional analyses, and are therefore not directly comparable.
Author Response
My only major comment relates to the interpretation of the results for genes involved in the biosynthesis of deoxynucleotides and amino acids. My understanding is that the chemical genomics screen was performed on M9 minimal medium. This medium would lack exogenous deoxynucleotides and amino acids. As such, mutations in the biosynthesis of these compounds are expected to be lethal both in the presence and absence of gallium - i.e., these mutants also should not grow on the control plate. Could the authors clarify why the deletion of these genes appeared to have an effect specifically in the presence of gallium, and why there were non-essential on the control plate?
- This is a good comment and we have elaborated on this in the beginning of the discussion once several experiments to assess the growth of these mutants on M9 media were performed. We determined that the presence of residual nutrients originating from the LB plates account for the ability of the mutants to grow. More information can be found in the revised paper in this regard.
Line 72: Cepacia should be lower case
- This has been changed
Lines 214-242: I found this section to be difficult to follow, and I suggest the authors attempt to rewrite it to improve clarity. I think the confusion arose, at least in part, as the discussion appears to switch back and forth between two distinct functional analyses (Figure 2 [EcoCyc categories] and 3 [DAVID categories]). Clarifying which analyses they are referring to at a given time may help improve the flow of this section.
- We have modified this section to help the flow i.e. not switching back and forth
Line 269-270: I do not follow the logic of this sentence. If TolC exports enterobactin from the periplasm to the external environment, would it not be expected to export unbounded enterobactin (so that it can then scavenge for iron). Why do the authors expect that a TolC mutant would therefore result in the accumulation of Ga-bounded enterobactin in the periplasm?
- This has been further explained in the manuscript
- We have concluded this since the presence of less enterobactin the extracellular pace provides fewer Ga targets, therefore the chance of Ga-enterobactin complexes traveling into the cell decreases.
Line 293: gloB should be in italics.
- This has been corrected
Lines 454-456: This comment is related to my comment about Lines 214-242. The first sentence describes functional mapping with the EcoCyc categories, while the second sentence switches to describing results of the annotation with DAVID categories. However, the authors do not specify that these two sentences refer to distinct functional analyses, and are therefore not directly comparable.
- We have made this more clear in our revision.
Round 2
Reviewer 1 Report
The manuscript was improved. Nevertheless, I feel that the WT MIC should be shown (the reader should know if the results were obtained from a GA- sensitive or resistant strain), and Figure 4 and 5 captions should show the name of the genes (the caption should include enough information to be interpreted without reading the supplementary table).